# Identity-Guided Spatial Attention for Vehicle Re-Identification

**DOI:** 10.3390/s23115152

**Published:** 2023-05-28

**Authors:** Kai Lv, Sheng Han, Youfang Lin

**Affiliations:** Beijing Key Laboratory of Traffic Data Analysisand Mining, School of Computer and Information Technology, Beijing Jiaotong University, Beijing 100044, China; lvkai@bjtu.edu.cn (K.L.); shhan@bjtu.edu.cn (S.H.)

**Keywords:** vehicle re-identification, deep learning, machine learning, attention mechanism, vehicle details

## Abstract

In vehicle re-identification, identifying a specific vehicle from a large image dataset is challenging due to occlusion and complex backgrounds. Deep models struggle to identify vehicles accurately when critical details are occluded or the background is distracting. To mitigate the impact of these noisy factors, we propose Identity-guided Spatial Attention (ISA) to extract more beneficial details for vehicle re-identification. Our approach begins by visualizing the high activation regions of a strong baseline method and identifying noisy objects involved during training. ISA generates an attention map to mask most discriminative areas, without the need for manual annotation. Finally, the ISA map refines the embedding feature in an end-to-end manner to improve vehicle re-identification accuracy. Visualization experiments demonstrate ISA’s ability to capture nearly all vehicle details, while results on three vehicle re-identification datasets show that our method outperforms state-of-the-art approaches.

## 1. Introduction

This paper investigates the challenge of identifying a specific vehicle from a vast image gallery database, known as vehicle re-identification [1,2,3,4,5]. The accuracy of this task relies heavily on the use of deep learning techniques [6,7,8,9] and computational resources, which are constrained by the availability of large-scale datasets. However, practical datasets of vehicle images obtained from traffic cameras are often riddled with noise, including occlusion and background artifacts. These nuisances adversely affect the efficacy of deep learning models and significantly compromise the accuracy of the re-identification task. Therefore, effective training on noisy vehicle images is crucial for achieving reliable vehicle re-identification results.

In vehicle re-identification, methods primarily rely on visual appearance as a means of distinguishing vehicles, rather than the license plate. Despite being the most distinctive characteristic of a vehicle, the license plate may prove to be ambiguous and untrustworthy under certain circumstances. Firstly, the low resolution of the license plate image captured alongside the vehicle may render it difficult to decipher the characters accurately. Secondly, the license plate can be obscured or even falsified, leading to its unreliability as a reliable identifier. Therefore, visual appearance-based approaches have been adopted in the majority of existing literature to tackle the challenge of vehicle re-identification.

Vehicle details are important in enabling the identification of distinct vehicles. As such, vehicle re-identification methods should concentrate on differentiating vehicles from other sources of noise, such as backgrounds or occluders. To achieve this, some existing methods employ attention mechanisms to emphasize vehicle-specific details. For example, He et al. [10] leverage a local region detector to identify areas such as windows, lights and brand logos. Similarly, Meng et al. [11] segment a vehicle into four distinct views and implement a shared visible attention mechanism to extract view-aware features. Wang et al. [12] propose a method that detects vehicle keypoints and extracts features from local regions. These approaches [10,11,12] achieve competitive accuracy in vehicle re-identification, underscoring the significance of vehicle details in extracting discriminative features from noisy data.

In order to visualize the regions to which discriminative features attend, as depicted in Figure 1, we employ the Class Activation Map (CAM) technique [13], which facilitates the generation of activation maps for input images. The method presented in [14] serves as a robust baseline for re-identification tasks, surpassing the performance of numerous vehicle re-identification approaches [10,11,12]. In this paper, our objective is to examine the attention map of the aforementioned baseline in order to discern the origins of the extracted features. It is important to note that the proposed method builds upon the foundation laid by the baseline described in [14].

The baseline method exhibits two primary issues with regard to attention. Firstly, as illustrated in the first two examples of Figure 1, the baseline extracts features from several patches that are unrelated to the vehicle. The activation regions encompass not only vehicle components but also various other objects, such as guardrails and green belts. Clearly, these objects are not integral to the vehicle and introduce significant noise into the feature embedding. Secondly, as demonstrated in the last two examples of Figure 1, the baseline may yield fragmented feature attention. In such instances, the baseline fails to concentrate on the vehicle’s appearance, instead directing attention towards the majority of the background. However, the background serves as a source of noise for vehicle re-identification and should not be incorporated into the generation of feature embeddings.

We propose the Identity-guided Spatial Attention (ISA) method to regulate the feature-extraction process, ensuring that the features are focused on vehicle-specific details. Specifically, we propose Single Identity-guided Spatial Attention (SISA), which assigns importance scores to each class at the spatial level. Multiple SISAs are integrated to generate an ISA map, emphasizing the discriminative regions within the input image. Subsequently, an ISA module is incorporated into the vehicle re-identification framework in an end-to-end fashion. During this procedure, the neural network’s attention is enhanced, shifting from a few localized details to a more comprehensive coverage of discriminative regions.

ISA is capable of filtering out noisy factors, yielding more relevant and focused attention. As demonstrated in Figure 1, the attention maps reveal that the proposed method zeroes in on specific vehicle details, such as lights and wheels. Notably, the algorithm disregards noisy areas, including guardrails and green belts, which are deemed irrelevant. In contrast, the baseline method extracts features from these unrelated objects. Furthermore, experimental results suggest that the relevant and focused features contribute to improved re-identification accuracy.

Additionally, the proposed method can be readily implemented in an unsupervised manner. In contrast, to thoroughly exploit vehicle details, some approaches [10,11,12] incorporate supervised detection components, such as vehicle keypoint or region detection. This inevitably leads to increased computational cost and can adversely impact the method’s applicability. In the present study, no supplementary manual annotation is employed for the identification of discriminative regions. Moreover, visualized results substantiate the accuracy of discriminative regions identified via our method.

In summary, the primary contributions of this paper are as follows:We introduce a spatial attention method to eliminate noisy factors and concentrate more on vehicle-specific details. An attention map is generated to highlight the discriminative regions of the input image.We propose an ISA module that leverages the ISA map to produce an attention matrix. The feature maps are refined by the attention weight, resulting in the acquisition of robust features.Distinguished from previous attention methods, ISA constitutes an unsupervised technique, necessitating no supplementary manual annotation and readily adaptable to other vehicle re-identification frameworks.

## 2. Related Work

### 2.1. Vehicle Re-Identification

To achieve a discriminative feature representation for vehicle re-identification, previous methods have utilized vehicle viewpoint or orientation cues. However, since vehicles are captured by cameras from various angles, images of identical cars from different orientations can vary significantly, making it challenging to identify similar vehicles of the same orientation. To address this issue, Wang et al. [12] propose a novel framework that includes orientation-invariant feature embedding and spatial-temporal regularization. Additionally, they utilize a key point regressor to obtain vehicle key points, which can distinguish similar cars based on subtle differences.

Another approach to vehicle re-identification is to learn viewpoint-aware deep metrics, as demonstrated by Chu et al. [15], who use a two-branch network. Zhou and Shao [16] adopt an adversarial training architecture with a viewpoint-aware attention model to infer multi-view features from single-view input. Zhou et al. [17] address the uncertainty of different vehicle orientations by using an LSTM to model transformations across continuous orientation variations of the same vehicle. Zhu et al. [18] propose training re-identification models for vehicle, orientation and camera separately. They then penalize the final similarity between testing images based on orientation and camera similarity.

Previous works have also exploited local details and regions to address vehicle re-identification problems [10,19,20,21]. Meng et al. [11] use a parser to segment four views of a vehicle and propose a parsing-based view-aware embedding network to generate fine-grained representations. He et al. [10] address the near-duplicate problem in vehicle re-identification by proposing a part-regularized approach that enhances local features. Liu et al. [22] introduce a multi-branch model that learns global and regional features simultaneously. They use adaptive ratio weights of regional features for the fusion process and propose a Group–Group loss to optimize the distance within and across vehicle image groups. Shen et al. [23] propose a two-stage framework that incorporates important visual-spatial-temporal path information for regularization. Khorramshahi et al. [24] present a self-supervised attention approach for vehicle re-identification to extract vehicle-specific discriminative features. Liu et al. [11] propose PVEN for view-aware feature alignment and enhancement in vehicle ReID using parsing and attention mechanisms. PVEN parses vehicles into four views, aligns features via mask average pooling and enhances features using common-visible attention. PCRNet [25] also utilizes vehicle parsing to learn discriminative part-level features, model the correlation among vehicle parts and achieve precise part alignment for vehicle re-identification.

In this paper, we also utilize discriminative regions to improve re-identification accuracy, where a region mask is generated to highlight vehicle details without additional manual annotation.

### 2.2. Attention Methods

The attention mechanism was originally introduced in Natural Language Processing [26,27,28,29]. Bahdanau et al. [26] propose an extension to the encoder–decoder model that learns to align and translate jointly in machine translation. The model automatically searches for parts of a source sentence that are useful in predicting a target word, rather than deploying a hard segment. Luong et al. [30] introduce global and local attention mechanisms to neural machine translation, where global attention considers all source words and local attention focuses on a subset of source words at a time. Vaswani et al. [31] propose Transformer, which replaces the recurrent layers with multi-headed self-attention. In this way, the sequence transduction model is entirely based on the attention mechanism. Shaw et al. [32] extend the self-attention mechanism to consider relative positions or distances between sequence elements.

Recently, attention mechanisms have been widely used in image classification [33,34,35,36], semantic segmentation [37,38,39] and object recognition [40,41,42,43,44], due to their ability to boost the performance of deep neural networks. Hu et al. [45] focus on channel relationships and propose the Squeeze-and-Excitation block, which adaptively calibrates channel-wise feature responses by explicitly modeling interdependencies. Wang et al. [46] propose the Residual Attention Network by stacking attention modules that generate attention-aware features. Woo et al. [47] propose the convolutional block attention module, which utilizes attention-based feature refinement with two different modules: channel and spatial. Park et al. [48] present a new approach to enhancing the representation power of networks via a bottleneck attention module. Bello et al. [49] introduce a two-dimensional relative self-attention mechanism as a stand-alone computational primitive, considering a possible alternative to convolutions. Li et al. [50] present the Harmonious Attention Convolutional Neural Network for joint learning of attention selection and feature representations and introduce a cross-attention interaction mechanism. Based on predicted part quality scores, Wang et al. [51] propose an identity-aware attention module to highlight pixels of the target pedestrian and handle the occlusion between pedestrians with a coarse identity-aware feature. Liu et al. [52] propose the Multi-Task Attention Network for multi-task learning, where a task-specific attention module is designed for each task to allow for automatic learning of both task-shared and task-specific features.

In this paper, we realize the attention mechanism by recognizing discriminative areas during training, due to the importance of regional clues in re-identification.

## 3. Method

In this section, we describe the proposed Identity-guided Spatial Attention (ISA) for vehicle re-identification. Firstly, we describe the spatial attention map that identifies comprehensive discriminative regions of the vehicle in the input image. Secondly, we describe the composition of the MIA module. Finally, we illustrate the network framework that incorporates the spatial attention module.

### 3.1. Preliminaries

The objective of vehicle re-identification is to retrieve gallery images that correspond to the same identity as a given query image. To achieve this, a model is trained on a dataset of *N* vehicle images and their corresponding labels, denoted as ${<I_{i},y_{i}>}^{N}i=1$. During training, the model is trained using an image classification approach that includes a Fully Connected (FC) layer. At test time, the FC layer is removed and the model maps vehicle images to feature vectors. The similarity of two images is then calculated by measuring the distance between their corresponding feature vectors, using the cosine distance metric in this paper. The images in the gallery set are ranked based on their similarity to the query image and relevant metrics are computed for evaluation.

However, we notice that the model is vulnerable to noise interference, such as backgrounds or occlusions, as shown in Figure 1. In addition, some sample images exhibit scattered attention, which negatively affects feature extraction. To overcome these challenges, we propose a spatial attention mechanism that can be incorporated into existing re-identification networks. Specifically, our approach concentrates the model’s attention and filters out noise factors. Consequently, the feature vectors contain less noisy information, resulting in improved re-identification accuracy.

### 3.2. Framework

The model architecture is shown in Figure 2 and it can be trained end-to-end without requiring regional annotations for learning ISA. To obtain a more informative feature map, we utilize a ResNet-50 variant as the backbone, which strikes a balance between accuracy and efficiency for most re-identification algorithms. The last classification layer that was originally trained for ImageNet [53] is removed and the stride of the last pooling layer is set to 1.

To integrate the proposed spatial attention into the re-identification framework, we introduce an identity-guided spatial attention module, which is detailed in Section 3.3 and Section 3.4. The ISA module can be broken down into two steps. In the first step, the 3D feature map is passed through a global average pooling (GAP) layer and an FC layer within the ISA module, generating an attention map called the spatial attention map. This map highlights discriminative regions while compressing noisy factors and shares the same size as the input feature map along the height and width dimensions.

In the second step, the ISA module is used to weight the feature map through element-wise multiplication, resulting in a refined feature map. This refined feature map is then passed through a GAP layer and an FC layer. To optimize the model, we employ both cross-entropy loss and triplet loss. Cross-entropy loss treats the re-identification task as a classification problem and is calculated as follows:(1)$$L_{CE}=-\sum_{i=1}^{B}y_{i}^{T}\log{\hat{y}}_{i},$$
where *B* is the batch size during training, $y_{i}$ is a one-hot vector representing the ground truth label of the image $I_{i}$ and ${\hat{y}}_{i}$ is the corresponding probability distribution over all categories predicted via the model.

To further improve the model’s similarity learning, we utilize triplet loss which employs triplets of samples from different identities. Let Θ(·) denote the deep model that maps raw images to feature vectors. The triplet loss is computed as follows:(2)$$\begin{array}{c} L_{Tri}=\max\left(d\left(\Theta \left(I_{a}\right),\Theta \left(I_{p}\right)\right)-d\left(\Theta \left(I_{a}\right),\Theta \left(I_{n}\right)\right)\;+margin,0\right), \end{array}$$
where $I_{a}$ is an anchor image, $I_{p}$ is a positive instance with the same label as $I_{a}$ and $I_{n}$ is a negative instance with a different label. These three samples are all sampled from a training batch using a hard example mining strategy. The L2 distance metric function, d(·), is used to compute the distance between feature vectors and margin is added to encourage the loss backpropagation between positive and negative pairs.

The total loss of our method is a combination of the cross-entropy loss and the triplet loss, as follows:(3)$$L=L_{CE}+\alpha \cdot L_{Tri},$$
where α is a hyper-parameter that controls the weight ratio between the two losses.

### 3.3. Identity-Guided Spatial Attention

The process of the Identity-Guided Spatial Attention (ISA) module is shown in Figure 3. We first introduce the concept of the Single Identity-guided Spatial Attention (SISA) map, which indicates the importance of different regions within an image. Building upon SISA, we then describe the Identity-guided Spatial Attention (ISA) framework, which is obtained by aggregating multiple SISA maps. The ISA framework consists of the single ISA map for the target identity and the single ISA maps for the remaining identities. Finally, we utilize the obtained ISA map to extract features that focus on key details.

#### 3.3.1. Single Identity-guided Spatial Attention Map

In this part, we describe the generation of the Single Identity-guided Spatial Attention (SISA) map, which indicates the importance of different regions in an image. The proposed method is based on the ID-discriminative Embedding (IDE) model. We use the Global Average Pooling (GAP) layer and a fully connected layer of the IDE model. The feature map *F* after the GAP operation is denoted as $f=(f_{1},f_{2},\ldots ,f_{i},\ldots ,f_{c})$, where *c* is the number of channels and $f_{k}$ is calculated as follows:(4)$$f_{k}=\frac{1}{h\cdot w}\sum_{i=1}^{h}\sum_{j=1}^{w}F_{i,j,k},$$
where *h* and *w* denote the height and width of the feature map, respectively.

The resulting feature map from the GAP layer is then converted into a 1D tensor via the fully connected layer. Each value in the 1D tensor represents the probability of belonging to a particular category. This process can be expressed as:(5)p=softmax(W·f),
where *W* is an n×c weight parameter of the fully connected layer and *n* is the number of categories in the training set. The softmax(·) function normalizes the n×1 tensor to a probability distribution over the predicted classes. The output tensor *p* is then used in the loss functions. For the sake of simplicity, the activation function and normalization operation that are commonly used are omitted from the above description.

The spatial attention map generated via the ISA module for a specific identity label *l* is denoted as $M^{l}$, which is represented as an h×w tensor. It is calculated by taking the sum of each channel of the feature map *F*, multiplied by the corresponding weight in the fully connected parameter *W*. More specifically, the calculation is as follows:(6)$$M_{i,j}^{l}=\sum_{k=1}^{c}W_{l,k}\cdot F_{i,j,k},$$
where each spatial unit (i,j) of the feature map *F* is utilized. This results in an h×w tensor that indicates the importance of different regions for the specific identity *l*.

It is worth noting that for each class, the average value of the single ISA map is equal to the corresponding dimension of the softmax input. In other words, for identity *l*, we have:(7)$$\frac{1}{h\cdot w}\sum_{j=1}^{h}\sum_{k=1}^{w}M_{j,k}^{l}=g_{l}.$$

The input to the softmax function is denoted as *g* and it is obtained as the product of the learned weight matrix *W* and the feature vector *f*. Thus, $M_{i,j}^{l}$ represents the activation score of the position (i,j), which has a direct impact on the predicted probability of the *l*-th category. The higher the value of $M_{i,j}^{l}$, the greater the contribution of the corresponding position to the predicted probability of the *l*-th category, and conversely, the lower the value of $M_{i,j}^{l}$, the lower the contribution of the corresponding position to the predicted probability of the *l*-th category.

Furthermore, we introduce a hyperparameter *r* which represents the discriminative ratio and is used to control the size of the recognized discriminative regions. For a single ISA map, we can use a threshold to separate high and low activation levels and control the area of discriminative regions. Thus, *r* is defined as the ratio of the area of the discriminative regions to the entire image.

#### 3.3.2. Generating Identity-Guided Attention Map

The identity-guided attention map *M* consists of two parts: the single ISA map of identity gt$M^{gt}$ and the single ISA maps of the rest of the identities $M^{gt^{\prime }}$.

To improve the accuracy of discriminative region recognition, we first utilize a single ISA map of identity gt. Discriminative regions are indicated by high activation scores in $M^{gt}$. We define $M^{gt}$ as:(8)$$M^{gt}=M^{l},\;where\;l=gt.$$

Using a single spatial attention map is not always sufficient for accurate recognition of discriminative regions. To overcome this limitation, we propose to use multiple spatial attention maps. Specifically, we utilize all *n* identities except for the ground truth identity (gt). Then, we define the multi-identity attention map as a sum of all individual spatial attention maps, denoted as:(9)$$M^{gt^{\prime }}=ℜ\left(\sum_{l=1}^{n}M^{l}\right),\;where\;l\neq gt,$$
where *n* is the identity number when training. Different from $M^{gt}$, the discriminative regions of $M^{l^{\prime }}$ are indicated by low activation scores. Thus, we apply a reverse operation *ℜ* on the attention maps. This approach allows us to consider multiple categories to obtain a refined and robust spatial attention map, rather than relying on a single attention map of identity *gt*. To illustrate the effectiveness of the proposed approach, two examples of discriminative region recognition results are presented in Figure 4. It shows that some important patches such as lights and sunroof are highlighted.

In the field of re-identification, previous works have also explored the utilization of attention mechanisms. For instance, the method Quality-aware Part Models (QPM) [51] proposes an identity-aware attention module to emphasize pixels associated with the target pedestrian, thereby addressing occlusion issues between pedestrians using a coarse identity-aware feature. In contrast, our method focuses specifically on vehicle re-identification, tackling occlusions and extracting essential information from within the vehicles. Furthermore, while QPM derives attention information from the quality scores of various body parts, our approach leverages attention obtained through a classification task, specifically the class activation map. This distinction enables us to accentuate the discriminative regions within the vehicles for effective re-identification purposes.

### 3.4. Features with Identity-Guided Spatial Attention

In this section, we propose to use the ISA map to enhance the performance of vehicle re-identification. In widely used re-identification pipelines, the GAP layer is employed to generate a 1D embedding feature tensor by averaging all spatial units. However, this operation results in the loss of significant information at the spatial level. Therefore, we aim to focus more deeply on regions that contain abundant clues during training. To achieve this, we implement an attention weight matrix on the feature map before the GAP operation.

In Section 3.3, we present the ISA map for determining whether a region is discriminative or not, but it does not provide a specific importance score. To address this limitation, we introduce a parameter $w_{impact}$, which represents the attention weight of discriminative regions and is greater than zero. Additionally, we define an h×w tensor *A* as the attention weight matrix, where each element of *A* is equal to $w_{impact}$ or zero. The calculation procedure is expressed as follows:(10)$$F^{\prime }=F⊙\left(A+1\right),$$
where the ⊙ operation represents the Hadamard product. The resulting tensor $F^{\prime }$ represents the refined 3D feature tensor, which is then passed through the GAP layer. The weight matrix *A* can be easily obtained by binarizing the ISA map *M*.

## 4. Experimental Results

In this section, we evaluate our proposed method on three widely used vehicle re-identification datasets: VeRi-776 [55], VehicleID [56] and VERI-Wild [57]. We also describe the evaluation metrics used and implementation details.

VeRi-776 [55] dataset contains 51,035 images of 776 vehicles captured by 20 cameras. The training set consists of 37,778 images of 576 vehicles and the remaining 13,257 images of 200 vehicles are used for testing. The test set has 1678 images for the query set and the rest of the images are in the gallery set. The dataset presents challenges such as various viewpoints, complex backgrounds and different distances.

VehicleID [56] dataset has 221,763 images of 26,267 vehicles. The training set has 110,178 images of 13,134 vehicles and the remaining vehicle images form the test set. The test set has three subsets: small, medium and large, with 800, 1600 and 2400 vehicles, respectively. The dataset only contains two vehicle orientations—front and back—and there is only one matching image in the gallery for each query image.

VERI-Wild [57] dataset contains 416,314 images of 40,671 identities captured by 174 cameras over a month. The training set contains 277,797 images of 30,671 identities. The test set is split into three subsets: small, medium and large, with 3000, 5000 and 10,000 identities, respectively. The dataset involves complex backgrounds, various viewpoints and different illumination and weather conditions. There can be multiple matching images in the gallery set for a probe image.

We use mean Average Precision (mAP) and rank*k* accuracy as evaluation metrics, following [57]. The mAP metric is suitable for scenarios where each probe image has multiple matching gallery images. The average precision of one query image is calculated by:(11)$$\mathrm{AP}=\sum_{g=1}^{G}p\left(g\right)\Delta r\left(g\right),$$
where p(g) denotes the recognition accuracy of the first *g* gallery images in the retrieved list. When the *k*-th image contains the same identity as the query, Δr(g)=1. Then, mAP is defined as:(12)$$\mathrm{mAP}=\sum_{q=1}^{Q}AP\left(q\right),$$
where *Q* is the size of the query set. The rank*k* accuracy specifies the percentage of probe images that matched correctly with one of the top *k* images in the gallery set. Finally, we provide the implementation details of our proposed method.

### 4.1. Implementation Details

We adopt IBN-Net-50 [58] as the backbone network in our experiments, which is a variant of ResNet-50 [54]. IBN-Net-50 replaces part of the batch normalization with instance normalization, leading to more discriminative features with a negligible increase in computation cost. We set the stride of the last pooling layer to 1 to obtain larger feature maps. The proposed model is implemented using the Pytorch deep learning framework.

We resize input images to 256×256 during both training and testing phases. The batch size is set to 64 and the number of images belonging to one category in the batch is set based on the properties of the different datasets. The hyperparameter *r* is set to 0.5. For VeRi-776 [55], we randomly select 16 images of each vehicle identity in a batch, as each category has an average of 75.6 images. For VehicleID [56] and VERI-Wild [57], each class has 4 images in a batch. We train the proposed method based on a stable re-identification network for 120 epochs. We use SGD as the optimizer, with an initial learning rate of 0.01. We adopt a warm-up strategy for the first 10 epochs, where the learning rate is increased linearly from 0.0001 to 0.01. After the 30th epoch, the learning rate drops symmetrically to 7.7 $\times \;10^{-5}$ with a cosine annealing strategy. We employ a batch normalization layer following the GAP layer. During testing, we use the output of the batch normalization layer as the final embedding features for evaluation. Considering that the GPU in use is the NVIDIA RTX A4000 Graphics Card, the computation time for processing a batch of 128 images is 0.505 s. The required GPU memory is about 3000 M. It is important to note that these figures are specific to this GPU and may differ for other GPU models.

The discriminative ratio *r* of our method is set to 0.5. We apply random erasing and horizontal flipping during training for data augmentation. We follow the optimal parameter settings [59] designed for image classification tasks. Specifically, the parameters of random erasing are set to $s_{l}=0.02$, $s_{h}=0.4$ and $r_{1}=0.3$. The area of the erasing rectangle is randomly selected from the range $\left(s_{l},s_{r}\right)$ and $r_{1}$ controls the aspect ratio of the rectangular region. The probability of performing random erasing is set to 0.5. Horizontal flipping is also carried out with a 0.5 probability.

### 4.2. Parameters Analysis

The attention weight $w_{impact}$. The parameter $w_{impact}$ denotes the attention weight of obtained regions during training. We conduct experiments with $w_{impact}\in \left\{0.1,0.2,0.3\right\}$. Experimental results are shown in Table 1. It can be seen that when impact=0.2, ISA achieves the highest mAP regardless of the parameter *r*.

### 4.3. Comparison with State-of-the-Art

In order to evaluate the effectiveness of our proposed method, we compare it with several state-of-the-art methods including RAM [19], VAMI [16], PGAN [60], AAVER [61], PRN [10], PVEN [11] and FDA-Net [57]. RAM extracts both global and regional features using a region branch to focus on more details. PRN employs a key region detection model to allow the re-identification network to pay more attention to important areas, which are manually selected in advance. Similarly, PGAN detects more regions to achieve higher accuracy. PVEN introduces a parsing model to segment a vehicle into four views. However, note that PRN, PGAN and PVEN require extra detection or segmentation methods with corresponding manual annotations.

#### 4.3.1. Evaluation on VeRi-776

Table 2 shows the comparison of the proposed method with several state-of-the-art approaches, with +STR indicating spatiotemporal details utilized in the corresponding approaches. The results demonstrate that our method outperforms other methods, including those involving auxiliary data. The proposed method achieves an mAP of 0.821, which is higher than the other methods. Regarding rank1, the proposed method obtains a rank1 of 0.973, which outperforms most of the compared methods.

Our proposed method outperforms TransREID [62] and SAVER [24] by a significant margin among the methods that do not involve extra annotations. While SAVER achieves a competitive mAP (0.796) on VeRi-776, our proposed method outperforms it in the small set of VehicleID with a much higher rank1 value (0.871 vs. 0.799). These results demonstrate the effectiveness of our method.

Compared to state-of-the-art methods that use additional annotations, such as VARID [63], our method improves the mAP metric by 2.5%. VARID uses viewpoint labels in the training process. In contrast, our proposed method does not require any additional annotations, yet achieves higher mAP and rank1 values.

**Table 2 sensors-23-05152-t002:** We compare our method with several state-of-the-art vehicle re-identification approaches on the VeRi-776 dataset [55]. The evaluation criteria employed are mAP, rank1 and rank5. Methods that require extra annotations are denoted by §. Numbers in bold are the highest values.

Methods		mAP	rank1	rank5
LOMO [64]		0.096	0.253	0.465
BOW-CN [65]	§	0.122	0.339	0.536
EALN [66]	§	0.574	0.843	0.94
BIR [67]	§	0.707	0.904	0.97
RAM [19]	§	0.615	0.886	0.94
VAMI+STR [16]	§	0.613	0.859	0.918
GSTE [68]		0.594	0.962	0.989
VANet [15]	§	0.663	0.897	0.959
AAVER [61]	§	0.663	0.901	0.943
PRN [10]	§	0.743	0.943	0.987
PRF [5]	§	0.779	0.964	0.985
PCRNet [25]	§	0.786	0.954	0.984
TransREID [62]		0.782	0.965	-
TransREID+views [62]	§	0.796	0.970	0.984
PVEN [11]	§	0.795	0.956	0.984
PGAN [60]	§	0.793	0.965	0.983
SAVER [24]		0.796	0.964	0.986
VARID [63]	§	0.793	0.96	**0.992**
DFNet [69]	§	0.809	0.97	0.990
baseline		0.805	0.953	0.982
ISA (Ours)		**0.821**	**0.973**	0.990

#### 4.3.2. Evaluation on VehicleID

The performance comparison on VehicleID is presented in Table 3, where we compare the proposed method with state-of-the-art methods, including RAM [19], PRN [10] and PCRNet [25]. Our method, which integrates local and global features, achieves superior performance in terms of rank1 and rank5 compared to most previous works. While PCRNet [25] achieves good performance on VehicleID and outperforms our method in terms of rank1 on the large subset, it requires manually labeled parsing data to carefully exploit vehicle features. In summary, our proposed method shows competitive results on the VehicleID dataset without relying on auxiliary information.

#### 4.3.3. Evaluation on VERI-Wild

Table 4 presents the comparison results on the VERI-Wild dataset. The compared methods include GSTE [68], FDA-Net [57], SAVER [24] and PCRNet [25]. The proposed ISA achieves 0.830, 0.781 and 0.710 mAP on the small, medium and large subsets, respectively. GSTE [68] leverages spatio-temporal features to improve vehicle re-identification, but it still lags behind our method. FDA-Net [57] utilizes a multi-task framework to learn view-specific feature representations, but it requires extra annotations. SAVER [24] and PCRNet [25] achieve good performance by introducing additional semantic segmentation data, but our method outperforms both of them on all three subsets without extra annotations. These results demonstrate that the proposed method can effectively recognize vehicles in diverse scenarios and has great potential in practical applications.

## 5. Conclusions

In this paper, we proposed a novel Identity-guided Spatial Attention method for vehicle re-identification that exploits multiple spatial attention maps. Our method contributes by incorporating spatial attention for vehicle-specific details, introducing the Identity-guided Spatial Attention (ISA) module for feature refinement and offering an unsupervised technique that is widely applicable. It enables the model to focus on important vehicle-specific details, enhances feature representation through the ISA module and can be easily integrated into various vehicle re-identification frameworks without requiring additional manual annotation. The experimental results on three benchmark datasets demonstrate the superiority of the proposed method compared with state-of-the-art approaches. Our method achieves state-of-the-art performance on both VeRi-776 and VERI-Wild datasets and outperforms most of the compared methods on VehicleID dataset. Unlike many other approaches that rely on additional annotations to facilitate their training process, our method offers two significant advantages. Firstly, these extra annotations, such as key points, orientation or segmentation details, are expensive to obtain. By eliminating the need for such annotations, our method reduces the financial burden associated with data collection and annotation efforts. Secondly, the absence of these annotations also reduces the computational resources required during the training process. This efficiency makes our method more accessible and feasible for real-world deployment, as it can be trained and executed using fewer computational resources, ultimately increasing its practicality and wide applicability.

One limitation of this study is the reliance on a specific dataset for evaluation. While we have achieved promising results using this dataset, the generalizability of our method to other datasets and real-world scenarios may vary. Therefore, we will perform further validation on diverse datasets to establish the robustness and effectiveness of our approach in different practical scenarios.

## Figures and Tables

**Figure 1 sensors-23-05152-f001:**
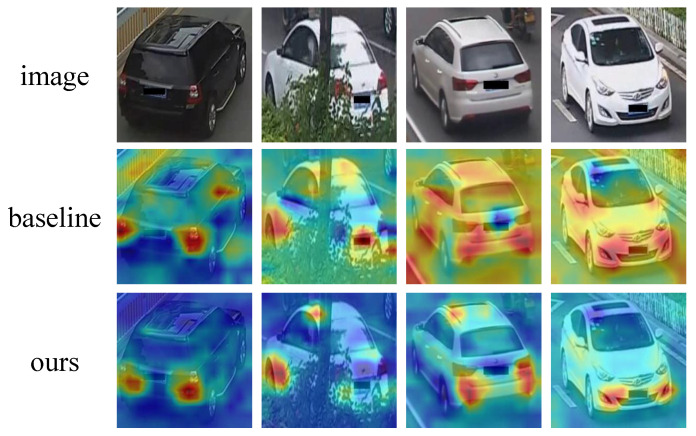
Illustrative examples of attention on vehicle images. We employ the activation map [13] to demonstrate the regions to which the features are directed. The baseline refers to the method proposed by Luo et al. [14]. Contrasting the baseline, which yields irrelevant and dispersed attention, our method exhibits relevant and focused attention.

**Figure 2 sensors-23-05152-f002:**
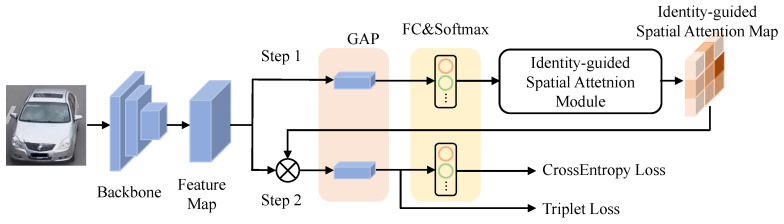
The structure of our network. The network includes ResNet-50 [54] as the backbone network, which generates a 3D feature map. The process can be divided into two steps. Firstly, the Identity-guided Spatial Attention (ISA) module is introduced to create an attention weight matrix. Secondly, the feature map is refined using the attention weight matrix through Hadamard product. Both steps include GAP and fully connected layers and employ cross-entropy and triplet loss for training.

**Figure 3 sensors-23-05152-f003:**
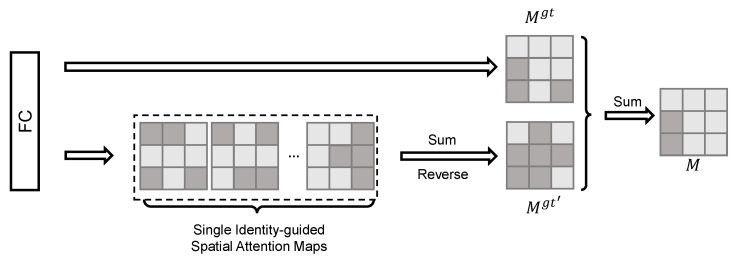
The process of the Identity-guided Spatial Attention (ISA) module, which aims to generate the identity-guided spatial attention map *M*, consists of two parts: the single ISA map $M^{gt}$ for the ground truth identity *gt* and the ISA map $M^{gt^{\prime }}$ for the other identities $l^{\prime }$.

**Figure 4 sensors-23-05152-f004:**
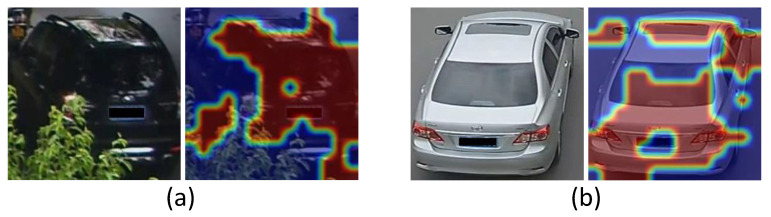
Two visualized spatial attention maps. Each pair consists of an original image and a corresponding image that highlights the recognized discriminative regions. Examples (**a**) and (**b**) can illustrate that our method will not focus on the occluder and the background, respectively.

**Table 1 sensors-23-05152-t001:** Parameter analysis of $w_{impact}$. Experiments are conducted on VeRi-776 [55].

$w_{\mathrm{impact}}$	mAP	rank1
0.1	0.793	0.957
0.2	0.814	0.970
0.3	0.824	0.978

**Table 3 sensors-23-05152-t003:** We compare our proposed method with state-of-the-art approaches on VehicleID [56], which is divided into three subsets: small, medium and large. Evaluation metrics include mAP, rank1 and rank5. The approaches that require extra annotations are denoted by §. Numbers in bold are the highest values.

Method		Small			Medium			Large		
		mAP	rank1	rank5	mAP	rank1	rank5	mAP	rank1	rank5
VAMI [16]	§	-	0.631	0.833	-	0.529	0.751	-	0.473	0.703
AAVER [61]	§	-	0.747	0.938	-	0.686	0.900	-	0.635	0.856
EALN [66]	§	0.775	0.751	0.881	0.742	0.718	0.839	0.71	0.693	0.814
RAM [19]	§	-	0.752	0.915	-	0.723	0.870	-	0.677	0.845
PRN [10]	§	-	0.784	0.923	-	0.750	0.883	-	0.742	0.864
SAVER [24]		-	0.799	0.952	-	0.776	0.911	-	0.753	0.883
PGAN [60]		-	-	-	-	-	-	0.839	0.778	0.921
TransReID [62]		-	0.823	0.961	-	-	-	-	-	-
PVEN [11]	§	-	0.847	0.970	-	0.806	0.945	-	0.778	0.920
PCRNet [25]	§	-	0.866	0.981	-	0.822	0.963	-	**0.804**	0.942
DDM [70]		0.823	0.757	0.905	0.802	0.743	0.889	0.785	0.731	0.853
VARID [63]	§	0.885	0.858	0.969	0.847	0.812	0.941	0.824	0.795	0.922
baseline		0.903	0.849	0.972	0.879	0.817	0.96	0.845	0.78	0.93
ISA (Ours)		**0.910**	**0.871**	**0.987**	**0.891**	**0.831**	**0.961**	**0.860**	0.791	**0.947**

**Table 4 sensors-23-05152-t004:** We present a comparison of the state-of-the-art vehicle re-identification methods on VERI-Wild [57], which has been divided into three subsets based on the number of vehicle instances. To evaluate the performance of the methods, multiple metrics such as mAP, rank1 and rank5 are used. Methods marked with § require extra annotations for training. Numbers in bold are the highest values.

Method		Small			Medium			Large		
		mAP	rank1	rank5	mAP	rank1	rank5	mAP	rank1	rank5
DRDL [56]		0.225	0.570	0.750	0.193	0.519	0.710	0.148	0.446	0.610
GSTE [68]		0.314	0.605	0.801	0.262	0.521	0.749	0.195	0.454	0.665
FDA-Net [57]	§	0.351	0.640	0.828	0.298	0.578	0.783	0.228	0.494	0.705
AAVER [61]		0.622	0.758	0.927	0.536	0.682	0.888	0.416	0.586	0.815
SAVER [24]		0.809	0.945	0.981	0.753	0.927	0.974	0.677	0.895	0.958
PCRNet [25]	§	0.812	0.925	-	0.753	0.893	-	0.671	0.85	-
VARID [63]	§	0.754	0.753	0.952	0.708	0.688	0.918	0.642	0.632	0.832
baseline		0.801	0.923	0.965	0.756	0.908	0.956	0.682	0.843	0.953
ISA (Ours)		**0.830**	**0.949**	**0.988**	**0.781**	**0.941**	**0.988**	**0.710**	**0.916**	**0.983**

## Data Availability

Not applicable.

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
