# Peer review of "Identity-Guided Spatial Attention for Vehicle Re-Identification"

_sensors, 2023, doi:10.3390/s23115152_

Round 1

Reviewer 1 Report

The paper is well written, experiments are made in more details.

The following aspects must be explained in more details:

-regarding: making it more practical and widely applicable in real-world scenarios: a more detailed explanation must be added: please explain more clearly how can you sustain that your method is applicable in real-world scenarios? what about computation time, memory needed (in fact all resources needed for making inference)

-how your proposed method is influenced by the speed of the car? How its performances are influenced by the speed?

-what are outdoor conditions (illumination, weather) used in the tested datasets? are only ideal conditions or some other conditions are tested?

Reviewer 2 Report

The authors' actual claim is what I'd like to know. It appears to be a use of a technique that has already been devised and its is applied on a different situation. It is advised to increase results for the readers' better comprehension. In comparison to the research previously suggested in "Quality-aware part models for occluded person re-identification," authors could discuss on their contribution. For better understanding, authors should provide a clear comparison and similarly follow the results of the comparison.  

It looks fine to me.

Reviewer 3 Report

Identity-guided Spatial Attention for Vehicle Re-identification

1.     I think the authors should add more to the keywords, based on the title and the proposed novelty of this research, the authors should try to include keywords that are relevant to this research. Using deep and machine learning is good, but the authors should be more creative and specific with their keywords.

2.     The primary contributions of this study should be summarized into 3.

3.     The authors should introduce a methodological framework showing the methodologies used in this study.

4.     The authors should include a limitation of the study and future recommendations of the research.

5.     The conclusion section still needs room for improvement in terms of explaining how the novelty of this research was achieved.

6.     Can I ask the authors why they used Identity-guided Spatial Attention (ISA) and not Single Identity-guided Spatial Attention (SISA) in the title of their manuscript?

7.     Another major question is how did the authors evaluate the methods using (table 3) mAP, rank1 and rank 5

The English language is moderate.

Round 2

Reviewer 1 Report

Since all my comments were addressed, I recommend to publish the paper.